# A Motion-Balanced Sensor Based on the Triboelectricity of Nano-Iron Suspension and Flexible Polymer

**DOI:** 10.3390/nano9050690

**Published:** 2019-05-02

**Authors:** Zhihua Wang, Songyi Yang, Shuang Miao, Qiongfeng Shi, Tianyiyi He, Chengkuo Lee

**Affiliations:** 1State Key Laboratory of Reliability and Intelligence of Electrical Equipment, Hebei University of Technology, Tianjin 300130, China; YangSongYi321@163.com; 2Key Laboratory of Electromagnetic Field and Electrical Apparatus Reliability of Hebei Province, Hebei University of Technology, Tianjin 300130, China; 3Experimental Training Center, Hebei University of Technology, Tianjin 300130, China; miaoshuang@hebut.edu.cn; 4Electrical and Computer Engineering, National University of Singapore, Singapore 117576, Singapore; eleshiq@nus.edu.sg (Q.S.); tianyiyi@u.nus.edu (T.H.)

**Keywords:** motion-balanced sensor, triboelectric, nano metal suspension, polymer

## Abstract

With the development of the Internet of Things and information technology, a large number of inexpensive sensors are needed to monitor the state of the object. A wide variety of sensors with a low cost can be made using the difference in charge attractiveness between flexible polymers and other materials. Compared to the two solid materials, a sensor made of a solid polymer-liquid has a large contact area and low friction. A motion-balanced sensor is presented based on the polytetrafluoroethene pipe and nano-iron suspension. The effect of the concentration and volume of the nano-iron suspension on the output voltage of the sensor is analyzed. The motion-balanced sensor can be used to measure the tilt angle of the object and there is a linear relationship between the output voltage and the tilt angle. A comparison test is performed to a commercial acceleration sensor with PZT-5. The test results show that the frequency characteristics and amplitude characteristics of the motion-balanced sensor are consistent with those of the acceleration sensor. The motion-balanced sensor can be used to determine the state of exercise such as walking, running, etc. The motion-balanced sensor has broad application prospects for monitoring the bridges and power towers balance, stroke patients’ health assessment, etc.

## 1. Introduction

Monitoring of object motion such as linear acceleration, angular acceleration and balance using sensor node is important in numerous applications including a robot, Internet of Things, elderly care and others. However, traditional motion monitoring sensors are complex to manufacture and costly. In contrast, flexible polymers can be large-scale manufactured easily at low cost [1]. Sensors using the law of charge attraction between a flexible polymer and other materials have become a hot topic of research. The main flexible polymers used in the literature are polydimethylsiloxane (PDMS), polytetrafluoroethene (PTFE), fluorinated ethylene propylene (FEP), polyethylene terephthalate (PET) and so on.

Most of these sensors are made of flexible polymers between two solids based on the law of charge attraction. Due to the difference in charge attraction, the surface of the two materials will accumulate positive and negative charges respectively when they are in contact with each other. Therefore, the voltage between the two materials will change as the contact surfaces of the two materials change. A variety of sensors can be made based on this principle. Tactile sensors can be made by the normal contact and separation of the PET [2]/PDMS [3,4,5] with other materials based on the potential difference generated. A self-powered motion sensor can detect by the random motions of a finger fabricated with FEP, PET and Al [6].

The application of liquid metals in the field of triboelectric sensor or generator has also received the attention of researchers because of their flexibility and stretchability. Liquid metal EGaIn electrodes are used for detecting objects of different geometry and generate triboelectric energy [7]. A triboelectric nanogenerator with liquid metal Galinstan as the electrode is studied for human body energy harvesting [8]. The nanogenerator works stably under various deformations such as stretching, folding and twisting. A liquid-metal-based triboelectric nanogenerator is developed with gallium and mercury [9]. Its output charge density is four to five times higher than that in the case if the electrode is a solid film. In addition, liquid metal can also be used as a triboelectric material. An acceleration sensor is made of a liquid metal mercury droplet and nanofiber-networked polyvinylidene fluoride film [10]. It can measure the vibration of mechanical equipment and human motion in real time.

Furthermore, the air gap between the two solid polymers affects their contact area, which in turn affects the amount of output charge and the stability of the signal [9]. Due to the fluidity of the liquid, the contact area between the liquid and the solid polymer is larger than that between the two solid polymers. Water [11,12], deionized water [13], and NaCl solutions are used for liquid-solid triboelectric generators to harvest energy from river or sea and triboelectric sensors also can measure ion concentration [14] or the flowing water velocity, position, reaction temperature, ethanol, and salt concentrations [9].

In this work, a motion-balanced sensor is designed and featured with nano metal suspension and flexible polymer. In order to make the liquid flow easily and possess a charged activity of the metal, a nano metal suspension is made by using nano-iron-powder, anhydrous ethanol and solid polyvinyl pyrrolidone (PVP). The triboelectric effect of the nano metal suspension is analyzed. PTFE film is used to make the arc-shaped pipe whose cross section is rectangular. Nano metal suspension can flow freely in the arc-shaped pipe. Acceleration and imbalance can cause the liquid to flow in the pipe and the contact area of the liquid and the PTFE film changes accordingly. Therefore, a corresponding voltage or charge signal is generated on the metal electrode due to the triboelectric effect. This sensor can be used to detect motion and balance and its shape can be changed depending on the application.

## 2. Materials and Methods 

### 2.1. Design and Fabrication

The motion-balanced sensor consists of polymer pipe, liquid and electrodes. The schematic drawing of the polymer pipe is shown in Figure 1a. Copper electrodes on the left and right sides are not connected, ensuring that the liquid flows back and forth with different potential outputs and the fabricated polymer pipe is shown in Figure 1b. The ratio of the actual object to the picture is 4:1 and the physical size is as follows: The length, width, and thickness of the upper PTFE film are 100 mm, 40 mm and 1 mm respectively. The upper PTFE film helps to maintain the shape of the pipe. The length, width and thickness of the lower PTFE film are 100 mm, 40 mm and 0.2 mm respectively. The two layers of PTFE films are separated by two PVC square columns, whereas the cross-section of the PVC column is a square with a side length of 2 mm. The bottom layer is two copper foil electrodes with a thickness 0.1 mm. Therefore, the rectangular polymer pipe fabricated has a height and a width of 2 mm and 30 mm, respectively. Each part is tightly bonded by a flexible polymer binder and the ends of the pipe are sealed with PVC.

In this paper, the nano-iron suspension is analyzed. An appropriate amount of such liquid is injected into the polymer pipe and the liquid accounts for one-third of the pipe. Nano-iron powder is made by laser sputtering and precipitation. The Fe powder is analyzed by scanning electron microscopy. As can be seen from Figure 1c, Fe powder has regular spherical structure, smooth surface. However, the size of the particle is not uniform enough. The number of particles with 50 nm, 50–500 nm, 500–800 nm and bigger than 800 nm are 30%, 14%, 16% and 40%, respectively. The number of particles smaller than 800 nm accounts for 60%. Some are linked into short-chain structure and have good dispersion. The position of the diffraction peak in the X-ray diffraction pattern of the nano-Fe powder shown in Figure 1d is compared with the X-ray diffraction file of Fe and the results are in good agreement. The cubic crystal structure of Fe can be determined and it is concluded that the main constituent of the phase in the sample is Fe. The (110) peak of Fe at 44.5 degrees is very sharp, indicating that the grain is relatively complete and there are no defects, which is consistent with the results of SEM scanning electron microscopy. No diffraction peak of iron oxides was observed in the figure, indicating that nano-iron powders oxidized slightly or not, which made Fe have better ability to lose charge in solution.

Solid, liquid or gas is dispersed in a liquid dispersion medium in the form of molecules or ions, particles and droplets under certain condition. Metal suspension can be made by evenly distributing metal particles in a solvent. The dispersant and dispersion media for each kind of metal particles are different. The metal suspension has more internal free charge and is more attractive to the charges when in contact with the flexible polymer resulting in a greater potential difference between the electrodes and power density, the instantaneous conversion efficiency is high. Therefore, it is beneficial to improve the sensitivity of the sensor [15]. Thus, we prepared a nano metal suspension to make the sensor.

Each metal has the same charge transfer tendency as it contacts the polymers [16]. Here, nano-iron suspension is made by using the nano-iron powder with a diameter of 500 nm, anhydrous ethanol and solid PVP. First, weigh 0.05 g nano-iron powder with an electronic balance. Then, mix it with 50 mL anhydrous ethanol and 2.288 g (4% wt.) solid PVP. Anhydrous ethanol is the solvent and solid PVP is a dispersant. The pH of the suspension is raised to 10 by adding caustic soda. Finally, the mixed liquid in a beaker is placed in an ultrasonic cleaner operating at a frequency of 40 kHz and a power of 300 W, afterwards, the liquid is shaken with ultrasonic waves for 20 min. Ultrasound can make the distribution of the nano-iron particles in the suspension more uniform. The suspension needs to be placed still for 24 h before use because small amounts of nano-iron powder that cannot be suspended need to be precipitated. The nano-iron suspension in the conical flask is shown in Figure 1e. Although ethanol is easily volatilized, since the sensor is in a sealed state, ethanol does not easily evaporate and can be used for a long time. The nano-iron powder added to the flasks 1, 2 and 3 is 0.05 g, 0.035 g and 0.02 g, respectively. The liquid 1 is saturated.

### 2.2. Working Principle

The working principle of the sensor is shown in Figure 1f. Inject 2 mL of liquid into the pipe so that the left and right ends of the liquid can correspond to the right and left sides of the electrodes, respectively. If the linear acceleration on the sensor changes, or the balance of the sensor is broken, the liquid is no longer in equilibrium but flows left or right and a potential difference is generated between the two electrodes. PTFE has a strong affinity for negative charges. PTFE film gains negative charges and the liquid acquires positive charges as the liquid flows through the pipe. PTFE has a steady state measurable surface charge. When the liquid flows over the surface of the material covered by the electrode, the electric field of the surface charge will be changed and it will produce electron movement between the electrodes. The strength of the material’s attractiveness to charge is one of the factors that influence the charge generation, but the ability of the liquid itself to lose its charge is also an important factor. Nano-iron powder metal suspension has better ability to lose charge than ordinary liquids due to their metal activity. If the liquid is in the middle of the pipe, the potential difference between two electrodes is 0.

For example, if the liquid flows to the left, the negative charge will collect on the PTFE film near the liquid. Then a large number of negative charges will collect on the left electrode. Due to the high electrical resistance of the PTFE film, the charge between the two electrodes cannot be neutralized in time. Therefore, the electrostatic balance is broken, and a potential difference is generated between the two electrodes. If a load is applied between the two electrodes, the negative charge will flow from the left electrode to the right electrode and the current will flow from the right electrode to the left electrode. Conversely, if the liquid flows to the right, the negative charge and current flow to the left and right respectively. Once the liquid stops moving and returns to the equilibrium position, the potential difference between the two electrodes returns to zero. However, if the liquid stops moving but is not in the equilibrium position, there will still be a certain potential difference between the two electrodes; the potential difference can be used to determine the tilt angle of the sensor.

In addition, the change of the potential difference is consistent with the change of the position of the liquid. If the sensor is attached to an object, the tilt or the linear acceleration of the object will cause the liquid to flow in the polymer pipe accordingly. Thereby, the equilibrium state and the motion law of the object can be determined by the potential difference between the two electrodes.

The relationship between the output voltage, the amount of charge and the capacitance value are analyzed below. Each electrode and liquid are one-third the length of the total length of the polymer pipe. Then the volume of the liquid V is:(1)$$V=\frac{abh}{3}.$$

Here, *a*, *b*, *h* are the length, width and height of the polymer pipe, respectively.

The amount of charge accumulated on the polymer pipe can be calculated from the volume of the liquid V. The pipe can be thought of as a capacitor after it has accumulated charge. The capacitance is *C*, the output voltage is U. The amount of charge with opposite signs on the electrodes and the generated potential with relative displacement (*z*(*t*)) of the surface are related [17]. Therefore, the voltage between two electrodes as follows:(2)$$U=\frac{Q}{C}-\frac{Qz\left(t\right)}{A\left(t\right)\varepsilon_{0}}+\frac{\sigma z\left(t\right)}{\varepsilon_{0}}.$$

Here, *Q* is the amount of charge accumulated on the electrodes. *Q* is the difference between *Q**_liquid_* and *Q_PTFE_*. *ε*_0_ is the electrical permittivity of vacuum and σ is the charge density. *A* is the contact area of the device. The contact area is a function of time due to the fluidity of the liquid.

## 3. Results and Discussion

A motion-balanced sensor is fabricated in Figure 1b. It is tested for balance sensing and motion sensing, respectively. Nano-iron suspension is injected into the polymer pipe for comparative analysis.

The output voltage is tested by a DPO3014 oscilloscope manufactured by Tektronix (Beaverton, OR, USA) with the input resistance 10 MΩ. In order to prevent power frequency interference, a filter circuit is used in the measurement to filter out the 50 Hz power frequency signal. A data acquisition card is used to collect the amount of charge on the electrodes.

### 3.1. Nano-Iron Suspension Test

When tilted to the left, the positive charge contained inside the liquid and the negative charge on the left PTFE surface cannot reach electrostatic equilibrium, which causes a right-to-left electron shift between the copper electrodes. Similarly, tilting to the right produces electrons moving in the opposite direction. The tilt surface test setup and charge distribution are as shown in Figure 2a,b. Both the suspension volume and the nano-iron powder concentration have a significant effect on the sensitivity of the sensor. The sensor is tested at a horizontal glass table top. Respectively 1 mL No. 1, 2, 3 nano-iron suspension is injected into the polymer pipe. First, the motion-balanced sensor is tilted to the left to the maximum angle. The output voltages corresponding to liquid 1, 2 and 3 are as shown in Figure 2c. Then, the motion-balanced sensor is tilted to the right to the maximum angle. The output voltages are shown in Figure 2d. Secondly, 2 mL of liquids No. 1, 2 and 3 are injected into the polymer pipe and the output voltages of the left and right tilts of the sensor are shown in Figure 2e,f, respectively.

It can be seen from Figure 2 that when the motion-balanced sensor is tilted leftmost, the voltage value is the lowest, which is a negative value; when the motion-balanced sensor is tilted rightmost, the voltage value is the highest, which is a positive value. Therefore, the tilt angle of the object can be judged by the output voltage value of the sensor. The comparison test of liquids 1, 2 and 3 shows that the suspension concentration has a significant effect on the experimental results. The higher the nano-iron powder content is, the higher the output voltages are. Therefore, it is advisable to use a saturated suspension for maximum sensitivity. The nano-iron suspension is saturated when 0.05 g of nano-iron powder is added to 50 mL of ethanol and liquid 1 is used in the subsequent tests.

The voltages in Figure 2e,f are significantly higher than the voltages corresponding to Figure 2c,d, indicating that the sensitivity on the condition of 2 mL liquid is greater than 1 mL liquid. This is also consistent with the theoretical analysis described above. Because in the equilibrium position, when the liquid is 2 mL, the suspension just covers the part without the electrode in the middle, no matter which direction the liquid flows, a potential difference will be generated on the electrode. Therefore, the sensitivity of the sensor at this condition is the highest. Conversely, if the liquid volume exceeds 2 mL, the liquid covers both the left and right electrodes when the sensor is tilted in one direction. In order to achieve electrostatic equilibrium, the number of electrons transferred between the left and right electrodes will be relatively reduced and the motion-balanced sensor output voltage will decrease. The output voltage of the three curves will drop by about 35 mV when adding 3 mL of liquid. This means that it is best to inject enough liquid into the sensor tube and not cover the electrode.

At the same time, we can see from Figure 2 that the output voltage will fluctuate greatly in the first few seconds during the deflection experiment. Due to the fluidity, the liquid will flow in one direction. However, the output voltage tends to be stable as the liquid is at rest.

### 3.2. Balance Sensing

When the sensor is tilted, the voltage between the two electrodes changes, so it can be used to detect whether the object is in equilibrium. Due to the structure and size limitations of the motion-balanced sensor, the maximum tilt angle to the left and right are both 20 degrees. Therefore, the range of tilt angle of the object that can be measured is [−20, 20] degrees. Here the negative angle sign is tilted to the left and the positive angle sign is tilted to the right. Firstly, test and analyze the dynamic characteristics of the motion-balanced sensor. Press the two ends of the sensor with fingers and rotate the sensor at a constant speed between −20° and 20°. The output voltage of the sensor during fast rotation and slow rotation is shown in Figure 3a,b respectively.

As can be seen from Figure 3, the amplitude and frequency of the voltage during rapid rotation are higher than the slow rotation due to the larger flow amplitude of the liquid during rapid rotation, indicating that the motion balance sensor can be used to detect the degree of deflection.

Secondly, the motion-balanced sensor measuring the static tilt angle. 2 mL nano-iron suspension is injected into the polymer pipe. The motion-balanced sensor is fixed to a flat plate and tilted at a specified angle. The motion-balanced sensor is tilted from the equilibrium position to the right by 10 degrees and hold for some seconds, then continue to tilt to the right to 20 degrees. The output voltage curve is shown in Figure 4a. And the tilting process is shown in Figure 4b. The output voltage rises from around 1200 mV to around 1450 mV and then rises to 2000 mV, then decreases to 1757 mV. The output voltage reaches a maximum and then falls. The test results show that there is a significant voltage changing process when the sensor is tilted from one angle to another, which is due to the fluidity of the liquid. Figure 4a shows that there is a direct relationship between the output voltage and the tilted angle. The variation of the voltage during the rotation at different angles is directly related to the rotational speed. The output voltage corresponding to each angle remains unchanged when the sensor maintains a steady state for a period of time. Thus, the output voltage can be used to determine the tilt angle of the object.

The motion-balanced sensor is slowly rotated from −20 degrees to 20 degrees. The output voltage is shown in Figure 4c and the sensor tilt process is shown in Figure 4d. Figure 4c shows that the output voltage of the sensor is the lowest at −20 degrees and the voltage gradually rises during the tilt to the right. Due to the flow of liquid during the rotation, the maximum output voltage is higher than the output voltage stabilized at 20 degrees when the sensor is tilted to 20 degrees.

The corresponding relationship between the output voltage and the tilt angle was analyzed. The voltages are measured every 5 degrees. The test result is shown in Figure 4e.

The tilt angle range is [−20, 20] degrees for the motion-balanced sensor. Figure 4e shows a linear relationship between the output voltage and the tilt angle. The angle sensitivity of the sensor is 3.65°/100 mV.

The output voltage in Figure 4 is higher than that in Figure 3. This is due to the fluidity of the liquid. When in a stationary state, the liquid can reach the position corresponding to the tilt angle of the sensor and in the continuous motion state, the deflection angle of the liquid is smaller than the deflection angle of the outer wall of the sensor. The above test shows that the sensor is capable of detecting the tilt angle of the object. Therefore, the motion-balanced sensor can be used for balance condition monitoring of buildings, transmission line towers, etc. The sensor is useful for giving an early warning of the danger of an object falling over. It can also be used for rehabilitation of stroke patients and assist doctors in assessing the patient’s condition.

### 3.3. Motion Sensing

The nano-iron suspension flows within the polymer pipe and creates a potential difference between the two electrodes when the sensor is excited by linear acceleration. Therefore, the motion state of the object can be measured by the proposed sensor. The sensor is fixed to a shell by bearings (Figure 5) instead of being directly attached to the object to increase the output voltage. The two bearings sandwich the central portion of the sensor from both sides of the shell but do not penetrate the side of the sensor so that it can be shaken around the centerline without being affected. The motion-balanced sensor data is amplified by a charge amplifier and then acquired by an oscilloscope. A single degree of freedom piezoelectric acceleration sensor is attached to the outer wall of the shell as shown in Figure 5. Acceleration sensor data is collected by the data acquisition card and sent to the computer. The sensitive component of this piezoelectric acceleration sensor is PZT-5, with sensitivity, measurement range and capacitance of 2.73 pC/m/s^2^, ±30,000 m/s^2^ and 1027 pF, respectively.

The shell is fixed on the tester’s body. The body will produce different vibration amplitudes and vibration frequencies when people are doing different exercises and the amplitude and frequency of the sensors connected to them will be different, so the output characteristics will also be different. We can analyze this output characteristic to understand the motion state of the human body. The output voltage of the motion-balanced sensor and commercial acceleration sensor is collected for analyzing the motion characteristics of the tester. The voltage amplitudes are quite different because the sensitivity of the two sensors is different. The voltage of different sensors is normalized for better comparative analysis. At the same time, the output voltage of the normalized motion-balanced sensor is subtracted by 1 and the two sensor output voltages after processing are placed in the same figure.

In the walking state, the processed voltage is shown in Figure 6a. It can be seen that the two voltages have a similar waveform law, but the fluid response speed is different from that of PZT-5 so that the phases of the two voltages are different. The motion-balanced sensor’s output voltage waveform is smoother. This is because the fluid itself has good low-frequency characteristics, which can filter out high-frequency noise interference in the environment and is more suitable for low-frequency motions detection, such as human motion etc. Fourier analysis is performed on the voltages of the acceleration sensor and the motion-balanced sensor and the amplitude-frequency characteristics are shown in Figure 6b. The amplitude-frequency characteristics show that the main energy is concentrated around 1 Hz, which directly reflects the frequency of human walking.

In the running state, results are shown in Figure 6c. Figure 6c shows that the two voltages have a similar waveform, but the phases are different. The voltage waveform of the motion-balanced sensor is also smoother. The amplitude-frequency characteristics are shown in Figure 6d. It can be seen that the main energy is concentrated around 2 Hz, which reflects the frequency of human running.

In addition, statistical analysis is performed on the test voltages of the motion-balanced sensor during running and walking. The sampling duration and number of data points are 10 s and 10,000, respectively. During walking and running situation, the average absolute values of the output voltage are 123.26 mV and 172.04 mV, respectively and the variances are 20,264.47 and 45,214.63, respectively. It is clear that the average absolute values and variance are significantly larger than walking in the running state. Therefore, the motion state of the object can be judged by analyzing the amplitude-frequency characteristics, the absolute average value, the variance and etc. of the motion-balanced sensor output signal.

The above analysis shows that the sensor can monitor balance and motion, and if the sensors are placed in three orthogonal directions, three-dimensional balance and motion can be measured. Moreover, the proposed sensor is inexpensive to manufacture. In addition, polymer pipes, nano-iron suspension and electrodes are easy to industrialize and large-scale applications.

The rate of precipitation of metal particles is primarily related to the ratio of dispersant and metal particle size. The dispersant PVP masses of suspensions ①, ② and ③ in Figure 7a are 1.144 g, 2.288 g and 3.432 g, respectively, with a ratio of 1:2:3. The precipitation is observed every 4 h. It can be seen that the suspension ① precipitates faster and the suspensions ② and ③ precipitates less after 4 h. Compared to suspension ②, the ③ will also have a little precipitation after 8 h. We continue to observe the precipitation in the next few days: suspensions ② will produce a little precipitation after 24 h, which has the slowest precipitation rate and then the precipitation rate will slow down afterwards. It can be seen that the precipitation of the iron powder suspension in the PVP saturated state is the slowest. After 24 h, it works best in the sensing field after removing sediment.

In order to analyze the effect of the metal particle size on the precipitation rate, suspensions ④ and ⑤, shown in Figure 7b, were prepared using iron powders having diameters of 400–1200 nm and 1000–1500 nm, respectively. The precipitation is observed every 4 h. It can be seen that both liquid precipitates are rare after 4 h. In addition, it is the same after 8 h. We continue to observe the precipitation in the next few days: both liquids will have a little precipitation after 24 h and the liquid No. 5 will produce a little more precipitation. It can be seen that the smaller the particles are, the slower the precipitation rate of the suspension is. It is better to use small particles when making a suspension.

Since the strong magnetic field has an effect on the characteristics of the nano-iron powder suspension, the sensor should not be used around high voltage lines and signal transmission towers. The suspension will produce a large amount of precipitation after 4 days and we needed to resume the sensor to make measurements with ultrasonic wave shaking. If the sensor is shaking for a long time, we don’t need to do any processing on the sensor.

## 4. Conclusions

This work developed a sensor based on the triboelectricity of nano-iron suspension and PTFE film for applications in balance and motion monitoring. The experimental results show that the nano-iron suspension has an obvious charge transfer when it is in contact with the PTFE film. The triboelectric sensor can be made of nano metal suspension instead of liquid metal which is expensive an easy to oxidize. The sensor can clearly perceive the tilt angle and motion of the object and can be used to judge the motion and balanced state of the object. The test results show that the saturated nano-iron suspension has the best output effect and if the particle is smaller, the suspension will be more stable. There is a linear relationship between the peak value of voltage and tilted angle, and the sensitivity of the angle is 3.65°/100 mV. By comparing to a commercial acceleration sensor, it shows that the motion-balanced sensor can be used for motion monitoring. Due to the flow characteristics of the liquid, the sensor is more suitable for measuring low-frequency motion and the voltage waveform measured by the motion-balanced sensor is smoother than the PZT-5 acceleration sensor.

## Figures and Tables

**Figure 1 nanomaterials-09-00690-f001:**
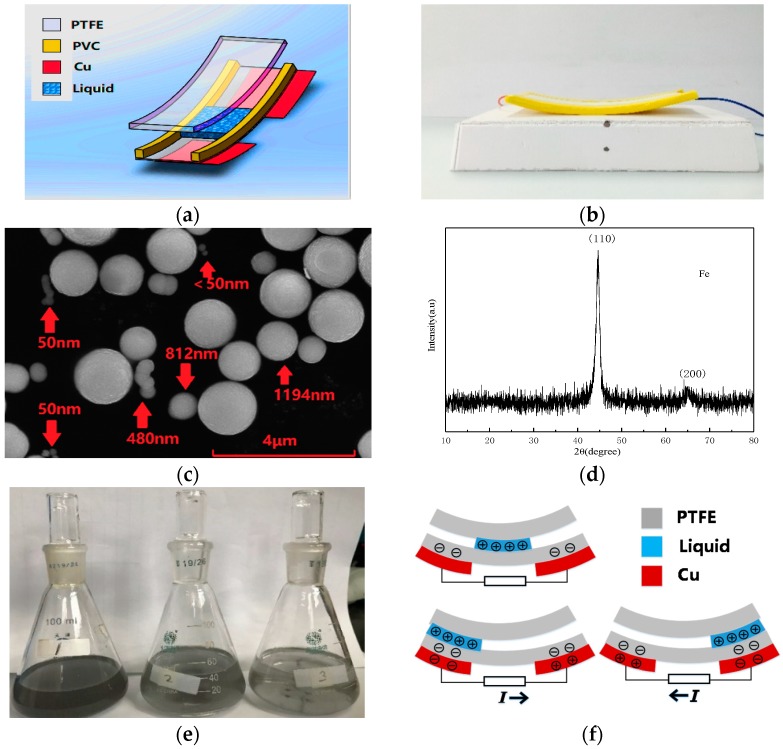
Schematic of the motion-balanced sensor (**a**), photograph of the fabricated sensor (**b**), SEM image of the Fe powder (**c**), XRD patterns of Fe powder (**d**), the nano-iron suspension in the conical flask (**e**), operation principle of the motion-balanced sensor (**f**).

**Figure 2 nanomaterials-09-00690-f002:**
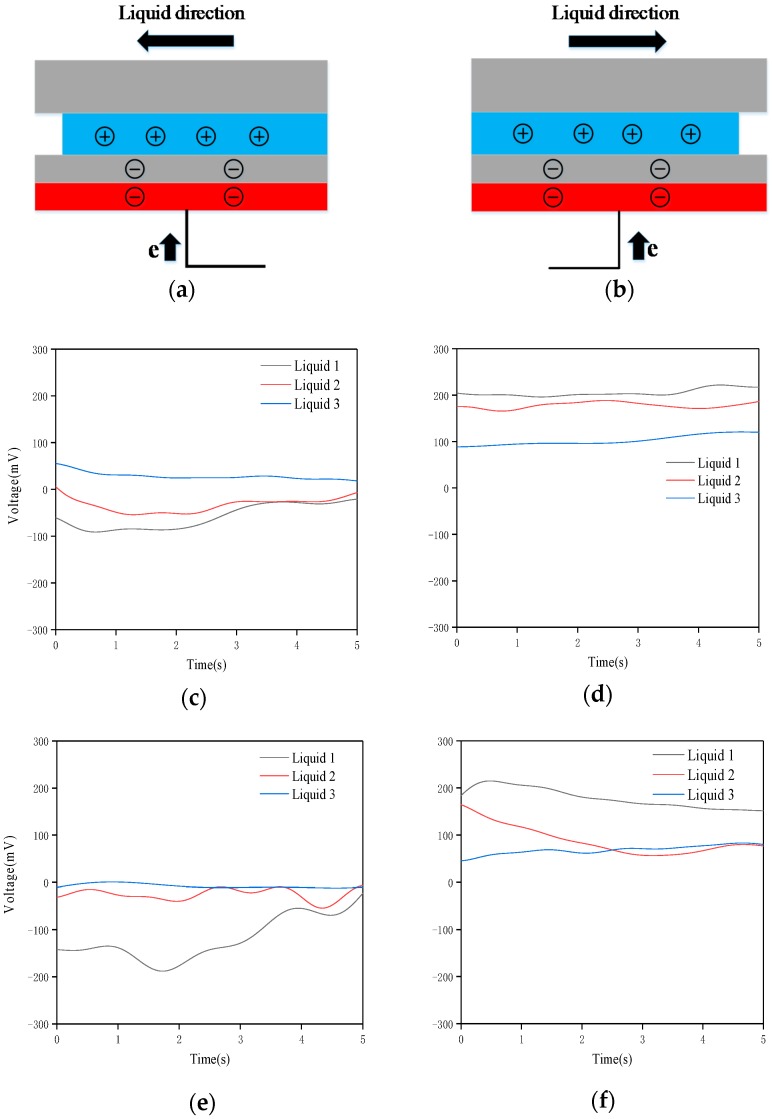
Tilt surface test setup illustration (**a**) and charge distribution description (**b**) The output voltages of the sensors with 1 mL liquid tilt to leftmost (**c**) and rightmost (**d**), with 2 mL liquid tilt to leftmost (**e**) and rightmost (**f**).

**Figure 3 nanomaterials-09-00690-f003:**
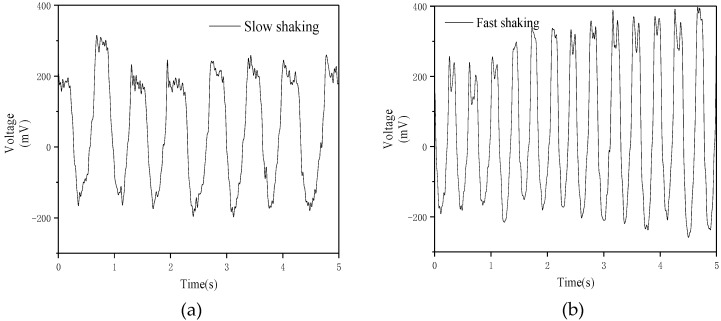
The output voltage when the motion-balanced sensor rotates slowly (**a**) and rapidly (**b**).

**Figure 4 nanomaterials-09-00690-f004:**
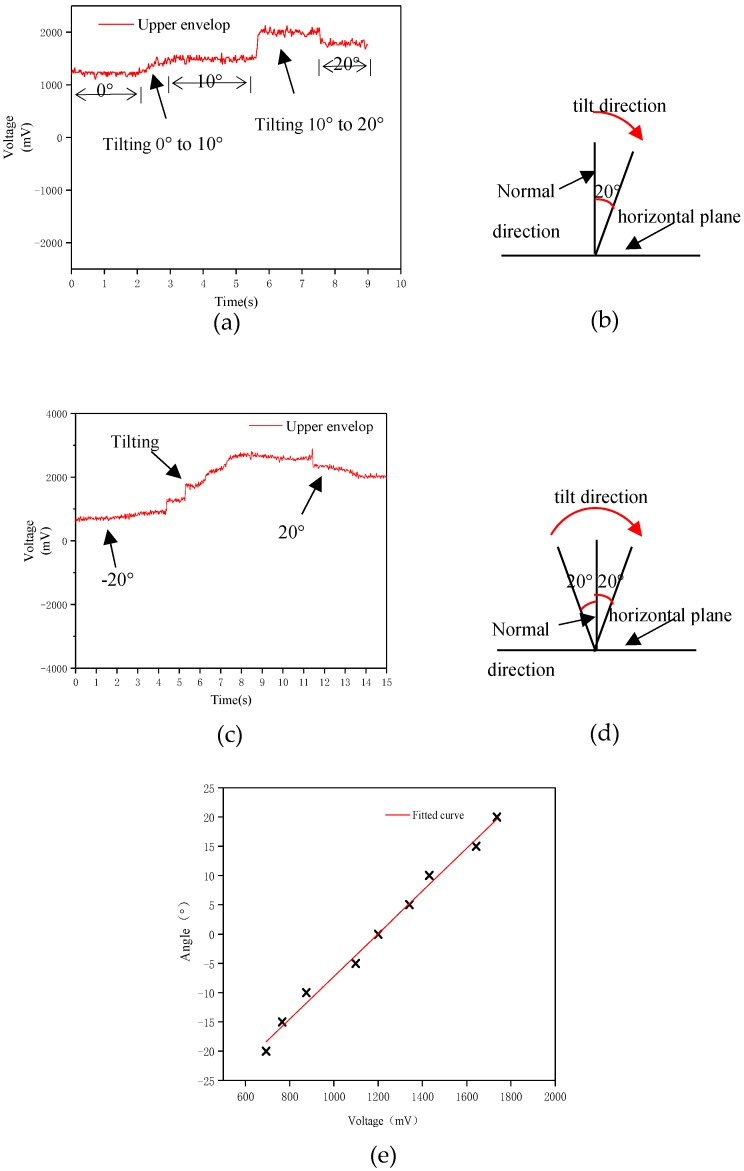
Output voltage of the motion-balanced sensor tilt 0–10–20 degrees (**a**) and tilt direction (**b**), output voltage of the sensor tilt from −20 to 20 degrees (**c**) and tilt direction (**d**), relationship curve between the output voltage and tilt angle (**e**).

**Figure 5 nanomaterials-09-00690-f005:**
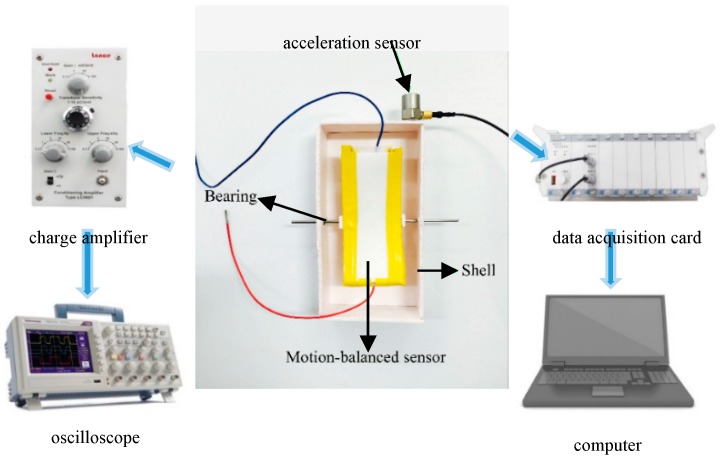
Motion test by motion-balanced sensor and commercial acceleration sensor.

**Figure 6 nanomaterials-09-00690-f006:**
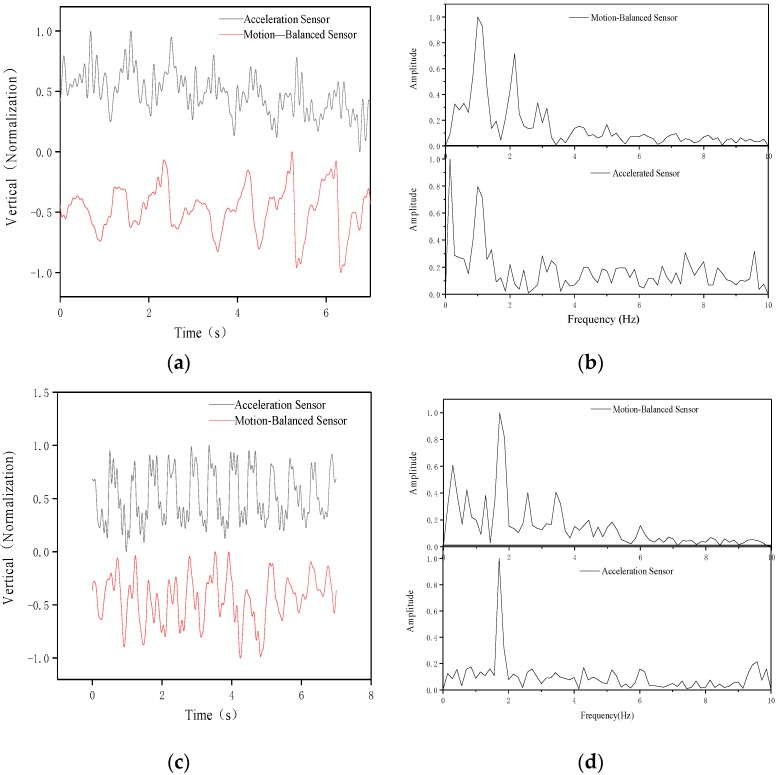
Normalized voltages (**a**) and amplitude-frequency characteristic curve (**b**) of the acceleration sensor and motion-balanced sensor while walking, normalized voltages (**c**) and amplitude-frequency characteristic curve (**d**) of the acceleration sensor and motion-balanced sensor while running.

**Figure 7 nanomaterials-09-00690-f007:**
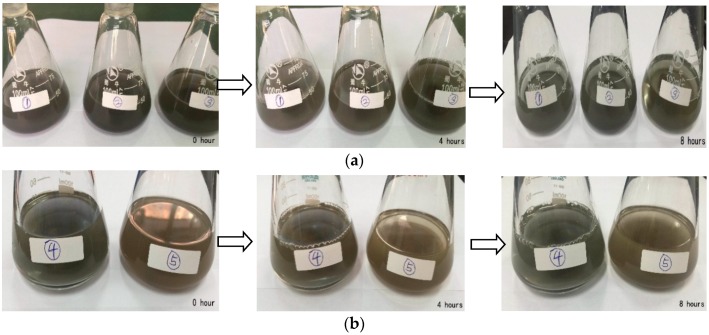
Sedimentation of iron powder suspension prepared with different concentrations of PVP (**a**) and sedimentation of suspensions with different particle size (**b**).

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
