# Peer review of "A Motion-Balanced Sensor Based on the Triboelectricity of Nano-Iron Suspension and Flexible Polymer"

_nanomaterials, 2019, doi:10.3390/nano9050690_

Round 1
Reviewer 1 Report
The authors have replied properly to my previous comments. A few more things need to be amended are listed here.
Page 3 line 121-123, the description was not clear.
The author should briefly explain what the number marks (1)-(3), and (4)-(5) stands for in the caption of Figure 7.
The charges in Figure 1(f) are still to small to recognize.
There will be no more comments from me before it could be accepted.
Author Response
Dear reviewer
Thank you for the comments concerning our manuscript entitled “A Motion-Balanced Sensor Based on the Triboelectricity of Nano Iron Suspension and Flexible Polymer” (Manuscript ID: nanomaterials-491218). Those comments are all valuable and very helpful for revising and improving our paper, as well as the important guiding significance to us. We have studied comments carefully and tried our best to revise. Revised text is marked with track changes. The main corrections and the responds to the comments are as flowing:
Point 1: Page 3 line 121-123, the description was not clear.
Response 1: Thank you for your valuable advice. We further described the statistical size analyses in line 121-123.
The number of particles smaller than 500 nm, 500-800 nm and bigger than 800 nm are 15, 4 and 15, respectively. The number of particles smaller than 800 nm accounts for 60%.
Point 2: The author should briefly explain what the number marks (1)-(3), and (4)-(5) stands for in the caption of Figure 7.
Response 2: Thank you for your valuable advice. We further explained what the number marks (1)-(3) stands for in the line 447-452 and what the number marks (4)-(5) stands for in the line 457-459.
The rate of precipitation of metal particles is primarily related to the ratio of dispersant and metal particle size. The dispersant PVP masses of suspensions 1, 2 and 3 in Figure 7(a) are 1.144g, 2.288g and 3.432g, respectively, with a ratio of 1:2:3.
In order to analyze the effect of the metal particle size on the precipitation rate, suspensions 4 and 5, shown in Figure 7(b), were prepared using iron powders having diameters of 400-1200 nm and 1000-1500 nm, respectively.
Point 3: The charges in Figure 1(f) are still too small to recognize.
Response 3: Thank you for your valuable advice. We enlarge the text and symbols in the Figure 1(f).
We thank you for your kind consideration of our article. Please do not hesitate to contact me directly if any further information is required. I look forward to hearing from you.
Yours sincerely
Zhihua Wang

Reviewer 2 Report
Authors have proven the points I pointed out in the last review. e.g. They added figure 7 about the sedimentation of the particles with respect to time, better than a single, stationary photo image. The article is maturated and started to become interesting with the changes they have made.
There is one important scientific mistake :
Authors claim: "The charge is present inside the material, and when the liquid flows over the surface of the material covered by the electrode, internal charges accumulate on the surface of the material."
This is a total mistake. Bandgap between HOMO and LUMO level in PTFE is around ~7 eV. Therefore, required actuation energy for electron transfer to the surface is too high (Only static electric field can't be enough). As a result, this claim is very wrong scientifically.
Polymers have a steady state measurable surface charge. When connected to an electrode, any static electric field change on the surface charge causes electron transfer through the metal which can be harvested by external circuits.
Author Response
Dear reviewer
Thank you for the comments concerning our manuscript entitled “A Motion-Balanced Sensor Based on the Triboelectricity of Nano Iron Suspension and Flexible Polymer” (Manuscript ID: nanomaterials-491218). Those comments are all valuable and very helpful for revising and improving our paper, as well as the important guiding significance to us. We have studied comments carefully and tried our best to revise. Revised text is marked with track changes. The main corrections and the responds to the comments are as flowing:
Point 1: There is one important scientific mistake: authors claim: "The charge is present inside the material, and when the liquid flows over the surface of the material covered by the electrode, internal charges accumulate on the surface of the material.
This is a total mistake. Bandgap between HOMO and LUMO level in PTFE is around ~7 eV. Therefore, required actuation energy for electron transfer to the surface is too high (Only static electric field can't be enough). As a result, this claim is very wrong scientifically.
Polymers have a steady state measurable surface charge. When connected to an electrode, any static electric field change on the surface charge causes electron transfer through the metal which can be harvested by external circuits.
Response 1: Thank you for your valuable advice. This explanation is indeed wrong, so it has been modified based on your comments in the line 158-161 with track changes.
“The charge is present inside the material and when the liquid flows over the surface of the material covered by the electrode, internal charges accumulate on the surface of the material.”——à “PTFE have a steady state measurable surface charge. When the liquid flows over the surface of the material covered by the electrode, the electric field of the surface charge will be changed and it will produce electron movement between the electrodes.”
We thank you for your kind consideration of our article. Please do not hesitate to contact me directly if any further information is required. I look forward to hearing from you.
Yours sincerely
Zhihua Wang

This manuscript is a resubmission of an earlier submission. The following is a list of the peer review reports and author responses from that submission.
Round 1
Reviewer 1 Report
A class of wearable and stretchable devices fabricated from thin films of aligned single-walled carbon nanotubes could be used to precisely monitor rapid and large-scale human motion with high durability, fast response and low creep. In the case of iron nano-particles it is necessary to prevent magnetic interactions. Iron nanoparticles tend to aggregate into micron-sized aggregates which sediment rather rapidly in high concentrated suspensions (for example 5 g/L), even in the absence of a magnetic field. The suspension stability of iron nanoparticles of different sizes, magnetic susceptibility over long time scales is the main problem.
This article includes a wide range of application fields from healing to homan movement monitoring.
The final article version would require:
1. Provide information on how and by what means the human movement information will be transmitted.
2. Provide information on how long measurements information will be reliable and how the monitoring results depends from the nano irron suspension sedimentation.
3. The manner in which the properties of nano iron suspesion can be restored to make measurements .
Reviewer 2 Report
The authors reported a motion-balanced sensor based on triboelectric nanogenerator (TENG) made from nano iron suspension. The design has severe flaw which cannot support the conclusion. The liquid is flow-able. If you tilt the device in a certain angle, the flow will result in varied voltage. At the same time, the authors used oscilloscope to measure, so the low resistance will induce leakage, which can also make voltage variation. That is why curves in Figure 2 has huge fluctuation. The voltage is only mV level, such high fluctuation make the result not reliable. Figure 3 shows the fast shaking can result in higher voltage, which is not good for sensing application. Figure 4 also shows highly non-reliable as well. So we cannot trust the conclusion that this device can be used for balance sensor. We cannot conclude the function of nano iron suspension as well since the tribocharge should be in the liquid. In all, this manuscript cannot be published in nanomaterials.
Reviewer 3 Report
The authors reported a balance and acceleration sensor based on triboelectricity and nano iron suspension. There is acceptable innovation in the research for being published in Nanomaterials. However, the acceptance should happen after the authors have properly addressed the following questions/mistakes.
1. In the first paragraph of 2.1, how were the parts assembled together. Were they glued to each other? Were the two ends of the pipe sealed after the suspension was injected?
2. Is there going to be evaporation problem to the ethanol which may affect the performance of the device, such as iron particle precipitation.
3. Page 3 line 132 the ‘conductively’ of the suspension was mentioned. What is this going to affect the performance since the suspension only plays as a triboelectric media but not an electrode?
4. Page 3 line 143, does ‘2ml (4%wt ) solid PVP’ refer to a solid or liquid? What is the solvent?
5. The captions and marks in some figures are too small, especially in Figure 1d and 1f.
6. Page 5 line 219-221. Is there experiment to support ‘if the liquid volume exceeds 2 ml, the output voltage will decrease.’
7. There has been a lot usage of the term ‘peak voltage’ in page 5 to 7. This is confusing since a lot are just continuous monitor of voltage, such as shown in Figure2 and Figure 4a, 4c.
8. In Figure 6, why it looks like the frequency of the signal from running is lower than that from walking? Should not it be the other way? This is also conflict to page 8 line 380 and line 385 where frequency of walking and running are mentioned. Besides, is this frequency directly reflect the frequency of the motion?
Reviewer 4 Report
The study is about developing a motion sensor based on triboelectricity. Authors use ion based (charged) liquid and oppositely charged flexible polymer. They plan to use this motion sensor for detecting the tilt angle of the objects. They compare the results using a PZT sensor. They claim results are matching with the commercial sensor. Finally, they have very ambitious application claims such as structural health monitoring, health assessment which is unrealistic.
This study is incomplete and not scientifically adequate. In the introduction and the main body text, a lot of points are missing and incorrect. The introduction is written scientifically false. Experiments are lacking discussion and control and they are not matching as claimed. What is claimed in the text and the plots/results aren’t matching. Although particle fabrication and characterization part seems okay, device fabrication and characterization part is very poor. This study, in my opinion, has no scientific merit and no merit to Nanomaterials.
1) In the title, abstract, and intro, authors claim that the motion sensor they developed is novel. First of all, this word should be excluded from the manuscript. Neither, liquid solid surface triboelectric effect nor triboelectric tilt sensor is novel. In addition, their geometric design isn’t also novel.
2) Introduction should be improved scientifically and quality wise. The English of the article is very poor. Article is disorganized in the introduction and overall.
3) Figure 1 a – Solidworks design looks very incomplete. There is a gap between Cu tapes and looks like there is no backing material. Transparency etc. should be changed.
4) Fig. 1b scale bar needed and material description as well.
5) Size and homogeneity of the iron particles are not clear ” … d particle size distribution between 482 and 1200 nanometers”. Fig. 1 c should be improved and a statistical size analyses should be included. What is the effect of particle size and phase of the iron particle to charge and output signal?
6) Photos of conical flask basically means nothing to readers.
7) Fig 1f. working principle description looks like correct but it is impossible to read it. They present like charges are inside the material. Charges are on the surface.
8) “The metal suspension has more internal free charge and is more attractive to the charge….” This is correct but should be scientifically proven by charge measurements or should referred to another study about internal charging.
9) “Inject a proper amount of liquid into 154 the pipe so that the…” What is the proper amount in numbers?
10) In working principle description, what i understand it there is only 3 states of the device: left, right and middle position. The resolution is very limited. O suggest authors to use more electrodes in series. Also, they can consider signal processing techniques for analyzing the signal characteristic vs contact area and flow rate. Here the main problem is, since the flow rate should also change the output signal intensity, they can only claim digitized state on and off. More mathematical work needed for sensor to be properly working.
11) Charge calculations are incomplete. There are many factors aren’t considered. Please check kanik et al, advanced materials 2015, main text and SI for the charging theory, models and experimental conditions.
12) Add a tilt surface test setup illustration and charge distribution description in Figure 2.
13) Figure 2- does show that you have DC voltage output? How did you obtain this while triboelectric effect can only give you instantaneous voltage peak with positive and negative parts?
14) More analysis needed for figure 3. Since the velocity also can change the output voltage, authors should have characterized frequency vs. signal at a certain angles, and angle vs signal at a resonant frequency. Lastly, multiplexed results of both frequency and angle. Otherwise, sensor can’t be used for meaningful sensing.
15) 3.65-degree sensitivity is incorrect and can’t be determined without the characterization at 14 and since frequency changes the signal intensity so sensitivity should be a function if the motion frequency rather than a single number.
16) Again, how do you obtain DV voltage in figure 4. It looks like capacitance measurement. Why measurement time is too long and you still have a straight line of signal rather than a peak?
17) Fig. 6 is a bit confusing. They show the Fourier of the both pzt and their sensors and claim they have the same characteristics. It looks correct that a person can take 12 steps in 7 seconds. But it is unclear why they only show 7 seconds of measurement? Why there is a shift in the signal amplitudes and shift in the baseline?
